# Does the Harvest Type Affect Olive Health? Influence of the Harvesting System and Storage Time on the Chemical, Volatile and Sensory Qualities of Extra Virgin Olive Oils

**DOI:** 10.3390/plants12223843

**Published:** 2023-11-14

**Authors:** Cosimo Taiti, Elisa Masi, Federica Flamminii, Carla Di Mattia, Stefano Mancuso, Elettra Marone

**Affiliations:** 1Department of Agri-Food and Environmental Science, University of Florence, Sesto Fiorentino, 50019 Firenze, Italy; elisa.masi@unifi.it (E.M.); stefano.mancuso@unifi.it (S.M.); 2Department of Innovative Technologies in Medicine and Dentistry, University “G. d’Annunzio” of Chieti-Pescara, Via dei Vestini, 66100 Chieti, Italy; federica.flamminii@unich.it; 3Department of Biosciences and Technologies for Agriculture, Food and Environment, University of Teramo, Via R. Balzarini 1, 64100 Teramo, Italy; cdimattia@unite.it (C.D.M.); emarone@unite.it (E.M.)

**Keywords:** harvesting system, olive cultivars, bruising effect, storage time, olive oil quality

## Abstract

With the aim of investigating the effect of bruising and its development during the postharvest time, olive fruits (Frantoio and Moraiolo), manually and mechanically harvested, were stored in climatic chambers at two different temperatures (5 °C and 18 °C) for five days. Visual observations highlighted changes in the olive peel with discoloration in the damaged areas and tissue bruising. Olive fruit polyphenols, volatile organic compounds (VOCs) and other oil quality parameters (phenolic content, free acidity and peroxide index) and sensory assessment were evaluated. Analyses were carried out on fruits and experimental extra virgin oils at harvesting and after 5 days of fruit storage. The results highlight that low-temperature storage (5 °C for 5 days) may contribute to the maintenance of high olive oil quality, and the quality of olives stored at room temperature drastically decreases after 5 days of storage. Moreover, mechanical harvesting, compared to manual harvesting, does not seem to affect the final oil quality, at least at harvesting, but seems to determine differences in the long-term storage period. Finally, the samples stored at 18 °C showed a quality deterioration with the development of sensorial defects.

## 1. Introduction

The cultivated olive (*Olea europaea* subsp. *europaea* var. *europaea* Green) is one of the most important crops in the Mediterranean area and represents a key element for the landscape configuration in traditional orchards. Since the beginning of the third millennium, EVOO (extra virgin olive oil) production has exceeded three million tons thanks to improved knowledge and harvesting technologies [1]. Harvesting is the final step in olive cultivation; this operation can affect the yield, olive fruit and oil quality, depending on the cultivar, and can markedly determine the income of the grower [2,3]. Until the 1980s, olives around the world were hand-harvested. Today, in Italy, mechanical harvesting is not a current practice. Indeed, notwithstanding the rapid advances in both trunk shaking and picker head technology, the development of mechanical harvesting is still slow if it is applied to old plants not structured for mechanical harvesting. In addition, the limited use of mechanical harvesting is linked to other factors such as area orography, farm size and lack of economic resources of the growers [4]. Thus, on the one hand, manual harvesting is characterized by low productivity and high costs [5]; on the other hand, mechanical harvesting stands offer a low-cost option and increase production quality [6]. Therefore, with the aim to increase the harvesting efficiency within the traditional olive orchards, a quick and suitable low-cost solution was to combine manual harvesting with hand-held shakers or shaker combs suitable for the olive harvest [7]. This improved efficiency can also damage the fruit (bruising on olive fruits) due to the impacts of the working organs or woody parts of the plant, which are more limited when fruits are harvested manually [8]. Unfortunately, olive oils obtained from damaged olives have high free acidity (FA), a high level of oxidation and a high content of some volatile acids (e.g., acetic or butyric) that develop off-flavors [9]. Moreover, the growing economic importance of premium EVOO compared to the standard EVOO productions further triggered new studies to better understand both the effect of the harvesting system and fruit storage. As per traditional practices, once harvested, the olives are stored at room temperature in olive harvest crates for a few days before milling. The maximum storage time is regulated by the law only for high-quality EVOO production, such as Protected Geographical Indication (PGI) and Protected Designation of Origin (PDO) (Table 1). Therefore, it seems important to understand (1) what happens over time within the bruising olives and (2) the effects of prolonged storage of healthy olives on olive oil quality.

The study of olive bruising due to mechanical harvesting has been undertaken for some time for table olives [10,11,12,13], while few studies were carried out about olives for oil production as well as on the effect of storage on bruised olives [14]. Therefore, the objectives of this study were (1) to evaluate the effect of mechanical harvesting by electric vibrating combs compared to the manual harvesting method; and (2) to verify the effect of the storage system (5 days of storage, 5 °C vs. 18 °C) on the harvested olives and the obtained oils. The study was carried out over two consecutive years and used fresh olives collected from two cultivars with different tolerance levels to bruising. Subsequently, we have produced some experimental oils to understand if and how olive damage and storage times could affect the final product quality.

## 2. Material and Methods

### 2.1. Plant Material and Climatic Conditions

Since the olive tree is often subjected to the phenomenon of the “alternate bearing”, that is, the tendency to produce a much greater than average crop in one year (year-on) and much lower than average crop in the following year (year-off), the trial was carried out for two consecutive years, 2018 (year-off) and 2019 (year-on) on olives collected from two different cultivars (“Frantoio” and “Moraiolo”), grown within a traditional olive orchard located at the “Fattoria di Macia” (Calenzano (FI), Italy, 43°53.23′52″ North Latitude, 11°09.23′04″ East Longitude, 87 m a.s.l.). This traditional orchard was constituted by self-rooted plants aged 25 years old, spaced 5.0 × 7.0 m, with a density of 286 trees/hectare. The “Frantoio” olive is a vigorous cultivar widespread in Italy, appreciated by growers and consumers for oil quality, but characterized by a thin fruit skin and a medium compact pulp. The “Moraiolo” olive is a local, medium-low vigorous cultivar also particularly appreciated for its oil quality [15], but it is characterized by a particular compactness of the fruit that makes it resistant to manipulation. The olive trees were ordinarily cultivated in a vase-shaped training system and had three long inclined branches with large open canopies. The trees were rainfed and grown with permanent inter-row grassing on a sedimentarious gipsy-arenaceous soil (pH = 7.2, organic matter 1%). Phytosanitary treatments were carried out to control pests according to organic cultivation rules. Since weather conditions largely affect the growth and fruit development as well as the relative obtained oils [16], climatic parameters were monitored daily during the two years of experiments. Indeed, climactic data (total rainfall, rainy days and temperature) were measured through a weather station (WatchDog 2425—SPECTRUM Technologies Inc., Thayer Court, Aurora, IL, USA) located inside the olive orchard.

### 2.2. Fruit Sampling, Harvesting, Storage and Oil Extraction

#### 2.2.1. Fruit Sampling and Harvesting

The trials were carried out during the ordinary harvesting campaign, which, in the Calenzano area, traditionally begins in the second half of October and continues until the end of November, depending on the fruit amount on the plants. The optimal harvesting times for these cultivars are different (the “Frantoio” fruits ordinarily ripen earlier than the “Moraiolo”), but as per the traditional custom related to the harvesting site, they were collected on 5 November 2018 (year-off) and 11 November 2019 (year-on), respectively. Indeed, the optimal harvesting time for “Frantoio” falls in the first half of November when the natural fruit drop has just begun, while “Moraiolo” is characterized by a late ripening. The following parameters were measured on the drupes of both cultivars at the harvesting time:

Resistance to Detachment (pool force) (RD)*:* This parameter was evaluated using a dynamometer (Correx, Switzerland) that measured the needed force (expressed in g) to break the fruit peduncle and detach it from the branch. To assess the RD, 200 healthy fruits were measured and randomly taken from the median external part of the canopy according to the four cardinal directions.

Fresh Weight (FW): This was evaluated on the same 200 fruits and expressed in g.

Ripening Index (RI)*:* This was determined according to Ferreira (1979) [17]; the ripening stage identification was measured on the same 200 fruits. The olive samples were classified according to their intensity color using an eight-class scale.

At harvesting time, the olives were submitted to:(1)Manual Harvesting: This was carried out by gently detaching the fruits from the plant manually and collecting them in woody baskets to avoid shocks and trauma with the fall; then, the olives were transferred to perforated plastic boxes, each containing about 20 kg of fruits.(2)Mechanical Harvesting: Hand-held olive harvesting by electric vibrating combs (Alice Top, by Campagnola) allows for the drupes to fall on nets around the tree. The fruits were subsequently transferred to perforated plastic boxes, each containing about 20 kg of fruits. As reported by Pegna et al. (2021) [18], the “Alice Top” has two opposed combs moving toward the other with 11 teeth each (6 long and 5 short, alternated), following an elliptical trace movement with an oscillation frequency of 19 Hz.

Subsequently, the harvested olives were split into different groups with the aim of testing the harvest systems (manual and mechanical) and two storage temperatures (5 and 18 °C). Three replicates per treatment were used for all the analyses.

#### 2.2.2. Fruit Storage Treatments

The harvested fruit samples were transported in plastic boxes to the laboratory within 2 h from harvesting and stored until the analyses were performed. Thus, the fruits were stored for five days in a dark climatic chamber (under a flow of humidified air) to avoid weight loss and the shriveling of fruits. Before the start of the storage time, the olive samples were split in two groups and then stored for five consecutive days (five sampling times) at 5 °C ± 1 in 2018 (year-off), and at two different temperatures (5 °C ± 1 and 18 °C ± 1) in 2019 (year-on). The temperature of 18 °C was chosen as standard storage condition for the olives at room temperature. Drupes were kept in separate olive boxes and for each sampling time (T1–T5), 1 kg of olives were taken from each storage condition and used for chemical and VOC analyses.

The time of storage was selected according to the Italian EVOO high-quality production rules (Table 1), while the storage temperature of 5 °C was selected to prevent a respiration increase, softening phenomena and fungi proliferation [19,20].

#### 2.2.3. Oil Extraction and Storage Treatments

During the year-on (2019/20 campaign), six different olive oil samples were obtained from different sampling times; in particular, two oils were obtained at T1 (manual and mechanical harvesting) and four oils at T5 (manual and mechanical harvesting; storage temperatures of 5 °C and 18 °C, respectively). All the experimental oil samples were blends obtained from 40 kg of fruits, 20 kg of “Frantoio” olives and 20 kg of “Moraiolo” olives, respectively. We decided to produce “blend samples” in order to simulate the traditional production process carried out in the area covered by our study. The olive oil samples were obtained using a laboratory mill (Olio Mio Baby 50; Toscana Enologica Mori, Firenze, Italy) equipped with a hammer crusher, horizontal kneader and two-phase decanters. Temperature (28 ± 2 °C) and time of malaxation (20 min) were standardized in order to minimize the variability of the procedure. The oil samples were put in dark glass bottles with a 0.25 L capacity and kept inside a dark climatic chamber (set at 15  ±  1 °C) until the chemical and sensory analyses were carried out.

### 2.3. Extraction of Phenolic Compounds from Olives

Olive fruit characterization was assessed for the 2019/20 harvest season. Three sampling times were considered for the assessment of total phenolic compounds (T1, T3 and T4), while two sampling times were used for the hydroxytyrosol, oleuropein, rutin and verbascoside evaluation. The extraction of phenolic compounds from the drupes was assessed according to Flamminii et al. (2021) [21] with slight modifications. Briefly, an aliquot of 1 g of olive pulp was mixed and homogenized with 5 mL of a MeOH/H_2_O 70:30 (*v*/*v*) solution for 1 min at 13,500 rpm (Ultra-Turrax model T25 Basic (Ika-Werke GmbH & Co., Staufen im Breisgau, Germany)). One milliliter of hexane was added and vortexed for 1 min. The mixture was centrifuged at 5300 rpm for 10 min. The organic fraction was discarded, while the hydroalcoholic subnatant was collected, filtered with 0.45 um nylon filters and stored at −40 °C until characterization. After proper dilution, the hydroalcoholic extracts were used for the Folin–Ciocalteu colorimetric assay adapted from Singleton and Rossi (1965) [22] and the ABTS radical cation discoloration assay. The results of total phenolic content (TPC) were expressed as mg Gallic Acid Equivalents (GAE) kg^−1^ dw, while antiradical activity was reported as mmol TEAC g^−1^ dw. The profiling of the main phenolic compounds of the extracts was assessed with a 1200 Agilent Series HPLC-DAD (Agilent Technologies, Milano, Italy) controlled with Agilent ChemStation for Windows (Agilent Technologies). The sample (10 μL) was injected into a C18 reversed-phase column, Kinetex 5 µm C18 100A 250 × 4.6 mm (Phenomenex, Bologna, Italy). The separation of phenolic compounds was carried out at a flow rate of 1 mL min^−1^ with a non-linear gradient from A (2% acetic acid solution) to B (acetonitrile). Gradient elution was as follows: from 10% to 20% B from 0 to 10 min, from 20% to 40% B from 10 to 15 min, from 40% to 80% B from 15 to 20 min, from 40% to 80% from 25 to 30 min, from 80% to 40% from 20 to 25 min from 40% to 10% from 25 to 30 min. The DAD acquisition range was set from 200 to 400 nm. Calibration curves were made with hydroxytyrosol, oleuropein, rutin and verbascoside, and the results were expressed as mg g^−1^ of olive dry weight.

### 2.4. Visual Bruising Assessment

The mechanical harvest effect, expressed as a bruising phenomenon (decolored and darkened areas on olive skin), was monitored by the same evaluator in both years of the trial for five sampling times and two different storage temperatures (5 °C and 18 °C). In particular, for each sampling time and cultivar, 100 fruits were observed, as per Jimenez-Jiménez et al. (2013a) [10], to evaluate the damage entity. In particular, to assess the Bruise Index by visual estimation (BI), the olives were split into different categories using a five-level scale of damage (where 0 = sound olives; 1 = slight damage; 2 = moderate damage; 3 = severe damage; and 4 = fruits with cut and mutilation).

### 2.5. Sensory Evaluation (SE) by Trained Panel

A total of six oil samples were subjected to SE following the rules proposed by the International Olive Council (IOC) (IOC/T.20/Doc. No 15/Rev. 10/2018 [23], Sensory Analysis of Olive Oil—Method for the Organoleptic Assessment of Virgin Olive Oil). At the DAGRI Department (University of Florence, Italy), eight panelists, trained according to the IOC rules (IOC/T.20/Doc. No 14/Rev. 4/2013 [24]), were recruited to classify the olive oil samples according to olfactory sensations, gustatory−retronasal sensations and finally, gustatory sensations. Judges were asked to identify the main positive and negative sensory attributes according to the IOC regulations. Thus, the overall sensory quality (aroma intensity, spiciness and bitterness) of each oil sample was evaluated by tasters according to a nine-point scale, where “1” highlights the poorest quality and “9” the best. Subsequently, based on their results, the oil samples were split into two categories: defective and not defective.

### 2.6. Instrumental Setup: VOC Detection from Olive Fruits and Olive Oils

The headspace volatile sampling for fresh olive fruits and extracted oil was conducted at 25  ±  1 °C (45–55% R.H.) inside a climactic chamber using a PTR-MS 8000 (Ionicon Analytik GmbH, Innsbruck, Austria). The tool was used in its standard configuration and using H_3_O^+^ as a reagent ion. Before starting the experiment, a multi-component standard gas containing methanol, acetaldehyde, acetone, and monoterpene mixture (Apel-Riemer Environmental Inc., Broomfield, CO, USA) was used to calibrate the tool. Each sample was analyzed in a randomized order, with a waiting time of 3 min between samples in order to avoid memory effects between one measure and another. Cleaning air (Zero Air Generator; Peak Scientific, Inchinnan, UK) was fluxed in the inlet between one measure and another. Raw data (expressed as counts per second, cps) were acquired with TofDaq software v. 183 (Tofwerk AG, Innsbruck, Switzerland) using a dead time of 20 ns for the Poisson correction and were subsequently converted to ppbv.

#### 2.6.1. Olive Fruits Sample Preparation and Analysis

Before the headspace analysis, the olives were selected based on their BI and then pitted through a manual olive stoner. Indeed, as reported by Masi et al. (2015) [25], the analysis was not applied to whole fruits because their volatile emissions are negligible. Subsequently, a sample of ~10 g of pitted olives (without stone) was placed inside a (3/4 L) glass jar with an apposite cap provided by two holes on two opposite sides that allowed for the connection through Teflon tubes with PTR and a zero air generator. For each cultivar and sampling time, three replicates were conducted, measuring the headspace mix for 3 min. For the analysis, the same instrumental setup reported by Masi et al. (2015) [25] was followed. Subsequently, all the interfering ions linked to the water cluster or with a value lower than <0.5 ppbv were discarded from the analysis.

#### 2.6.2. Olive Oils Sample Preparation and Analysis

The headspace analysis was done after the sensorial analysis using the same olive oil samples. The volatile fingerprints of six samples obtained in the 2019/20 season were obtained following the same setup and methodology reported by Taiti et al. (2022) [26]. Moreover, based on our previous results [26], we focused the analysis only on compounds previously identified as quality markers, such as m/z 45.033, m/z 59.049 and m/z 61.028 as off-flavor, while m/z 79.059, m/z 81.069 and m/z 99.080 as positive attributes (on-flavor).

### 2.7. Olive Oil Quality Parameter Analysis

For each olive oil, the samples were evaluated in terms of free acidity (%), peroxides (meq O_2_ kg^−1^) and total polyphenols (mg kg^−1^). Free Acidity (FA) and Peroxide Value (PV) were evaluated following the official methods described in EEC Reg. 2568/91, and subsequent modifications were made in terms of low levels of FA and PV, which highlighted a good quality of olive oils and vice versa [27]. Total phenols were determined by the Folin–Ciocalteau method [28]. For each oil sample, measurements were carried out in triplicate, and the values were averaged.

### 2.8. Statistical Data Analyses

Analyses of variance (ANOVA) were performed to determine if the different considered factors (years, cultivars, harvesting system, sampling times and storage temperatures) have a statistically significant effect on the Bruising Index (dependent variable). The separation of means was calculated by Fisher’s least significant difference (LSD) test. Computations were performed by Statgraphics Centurion XV v. 19.4.04.

## 3. Results and Discussion

### 3.1. Climatic Data and Fruits Characteristics

Table 2 reports the average monthly temperatures and precipitation data for two consecutive years. The summer temperature during the experimental work never exceeded 34 °C, while the average annual temperatures were 16.7 °C in 2018 and 16.2 °C in 2019, respectively. The total rainfall in the first year (2018) was 892.5 mm, with a peak in March of over 200 mm. However, the second year (2019) was particularly rainy (1023.3 mm), with two peaks in spring and late autumn (May, 136.4 mm; November, 327.8 mm), as reported in Table 2.

From a phenological point of view, it can be noted that the “Frantoio” always has higher FW and RI compared to the “Moraiolo” (Table 3). On the contrary, the RD was higher in both years in the “Moraiolo” (Table 3). Comparing the data obtained from different years, it can be seen that for both cultivars in the year-off (2018), the FW of the single fruit was higher, while the RD values were lower due to a more advanced ripening compared to 2019 (year-on). This behavior was more apparent in the “Moraiolo” olives compared to the “Frantoio”. On the contrary, the year-on (2019) showed a reduction in the FW of the single fruit and a considerable slowdown in the ripening processes, which determined a lower RI and a greater RD at the time of the tests. Thus, it appears that the phenomenon of “alternate bearing” together with weather conditions affected both the physiological development and the ripening process of the olive fruits. This result confirms a study by Mafrica et al. (2021) [16], which reported how the years characterized by higher rainfall showed a slower olive maturation process.

### 3.2. Visual Bruising Observations in Olive Fruit

Olives of both cultivars manually harvested showed only minor bruising damage; on the contrary, the olives harvested mechanically showed more remarkable damage on the skin due to the impact of the electric vibrating combs right from the first sampling time (Figure 1). Furthermore, the darkened areas in olives harvested by electric vibrating combs increase in postharvest conservation in both cultivars, while for olives manually harvested, no significant differences were detected both over time and between the studied cultivars (Figure 1). However, the cultivar that shows a better response to mechanical harvesting is “Moraiolo”, while “Frantoio” is more sensitive to the bruising effect (Figure 1). Our results agree with Sola-Guirado et al. (2022) [12], who reported the level of bruising is affected by cultivar and fruit ripening. Moreover, the bruising due to the harvest system observed here was the same reported by Jiménez et al. (2016) [29] on different olive cultivars such as “Manzanilla de Sevilla” and “Hojiblanca”. At T1, the darkened areas were much more visually noticeable in the not-fully-ripened fruits compared to the others. In addition, the color intensity of bruises increased during the time storage (Figure 1), confirming the previous results on table olives [29].

Indeed, the consequence of the physical impact is an enzymatic reaction that occurs inside the fruit and is responsible for visible tissue darkening [29,30]). The browning reaction resulting from mechanical injury is a widespread phenomenon in fruits, as is the positive effect of cold storage to reduce the browning effect and color changes in olives [31]. In addition, the brown color of the damaged area is mostly due to the oxidation of polyphenols by the enzyme polyphenol oxidase [32]. From Table 4, it appears that the main factor influencing the visual bruising is the harvesting system, which alone accounts for 42.9%, followed by the cultivar (23.9%). The year effect represents 10% of the total, and the harvesting time is 8%. The harvesting system × cultivar (2.4%) and harvesting system × year (5.4%) interactions are significant. It is interesting to note that the two interactions are linked to the characteristics of the cultivars, with the “Frantoio” having a more advanced maturation and is more susceptible to manipulation, which shows higher levels of visual bruising than that of the “Moraiolo”, which considered more resistant to manipulation and which, certainly, was harvested at a less advanced stage of maturation. The second interaction highlights the effect of the “year-on” compared to the “year-off”; this effect is equally mediated by the different degrees of ripeness of the harvested olives. In fact, “Year-off” determines a generalized advance in ripening, which is also reflected in manual harvesting with a higher percentage of damaged olives, even if slightly. In fact, to justify the data, it must be taken into account that to detach the fruits, some physical impact must be made to harvest them, which slightly damages the epidermal tissues.

As highlighted in Table 5, the factor that most affects the bruising phenomena is the cultivar (31.8%), followed by the harvesting system (26.4%). Likewise, the storage time action is also evident (16.7%), which determines a reduction of the bruising, which only appears starting from the third sampling.

### 3.3. Influence of Olive Cultivar, Harvesting System and Storage Temperature on Phenolics Content in Olive Fruits

The phenolic content of olive fruit depends greatly on biotic factors as well as the characteristics of the cultivation zone, the climatic conditions, the different techniques applied during the production and the harvest and the postharvest treatment [33]. The evolution of the total phenolic content of olive fruits collected during 2019 (year-on) and stored at different temperature conditions is reported in Table 6. Concerning the cultivar effect, at T1, the “Moraiolo” generally showed a higher TPC content than the “Frantoio,” with greater values in the case of drupes manually harvested and refrigerated at 5 °C. This is in line with other researchers who observed that “Moraiolo” itself is a cultivar richer in phenolic compounds with respect to “Frantoio” and “Leccino” [34]. During storage time, a decrease in the TPC was verified for “Moraiolo” in both the harvesting systems, with values that were significantly higher in the olives that were manually harvested (−5.9% to −17% for 5 °C and 18 °C, respectively), compared to the mechanically harvested samples (−1.1% to −3.9%, 5 °C and 18 °C, respectively). Considering that “Moraiolo” also showed the lowest RI value for the harvest in 2019, the above results could be related to the enzymatic activity of PPO, which is higher during the first months of fruit development but slows down during fruit maturation and ripening [35]. A slight increase was depicted for the “Frantoio” olives, mainly for mechanically drupes stored at 18 °C (+14.7%). This latest effect could be ascribed to both endogenous and/or microbial enzymatic activity, accelerated by the damage of cell structures happening during olives harvesting and storage at high temperatures, which affect the content of phenolic compounds. A similar behavior was also reported by Hbaieb and co-authors [36] that registered an increase in flavonoid content in Arbequina fruits during the second week of 20 °C storage. Furthermore, the TPC content could also be associated with altered physiological behavior during 2019, as previously discussed. Pearson correlations between TPC and antiradical activity values at T1 depicted a positive correlation mainly for drupes stored at a refrigerated temperature (0.96).

The influence of variety, harvesting system and storage conditions on the content of hydroxytyrosol, oleuropein, rutin and verbascoside in drupes was reported in Table 7. At T1, “Moraiolo” shows higher content of individual phenolic compounds than the “Frantoio”, confirming the trend observed for the TPC. During storage time, peculiar behaviors were depicted. Hydroxytyrosol (OHTyr) showed a general increase at 18 °C, mainly when drupes were manually harvested, especially for the “Frantoio” olives (+135.9%); contrarily, an opposite decrease in the phenolic alcohol was observed at 5 °C of −24.4% and −27.3% for the manually harvested “Frantoio” and “Moraiolo”, respectively. Oleuropein showed a marked increase when drupes were stored at 18 °C, mainly in manually harvested “Moraiolo” (+420.8%), which is likely associated with the high activity of b-glucosidase in green fruits [36]. Less pronounced was the increase for the olives stored at 5 °C, which lagged the decrease in enzyme activity and delayed olive ripening and tissue softening [37]. The flavonoid rutin showed a slight increase in “Moraiolo”, regardless of the temperature and harvest system, while an opposite trend was depicted in “Frantoio”, +57.5% and −3.8% at 18 °C and 5 °C, respectively, for the manual harvest and +0.5% and +50.6% at 18 °C and 5 °C, respectively, for the mechanical harvest. The general increase in OHTyr and Ole in the “Frantoio” and “Moraiolo” drupes at 18 °C could be ascribed either to oleuropein release, originally linked to different substrates in the fruit such as the polysaccharides, resulting from exogenous enzymes produced by micro-organisms during olive storage or to the compartmentalization of oleuropein and their degrading enzymes, as reported by Hbaieb et al. (2015) [36]. Concerning verbascoside, despite the high level in the fruits, mainly in the “Moraiolo”, when compared to the other phenolic compounds, a general reduction during storage was depicted except for mechanically harvested “Frantoio” at 5 °C (+40.9%). The peculiar evolution of phenolic compounds during storage is strictly linked to biotic factors (cultivar) but also to the agronomic factors that influenced the physiological status of the drupes, such as the harvesting system, as previously evidenced.

### 3.4. VOCs Results

#### 3.4.1. Influence of Olive Cultivar and Time Storage on VOC Emissions in Olive Fruits

Since the mechanical harvesting systems increased the level of bruising, olive storage at low temperatures could be a useful way to delay the degenerative effects inside the olives between the harvest and the crushing. Therefore, to understand the effect of bruising on fruits, we monitored for the first time how the volatile emissions from olives stored at 5 and 18 °C, respectively, changed over time in two different cultivars collected manually and mechanically by electric vibrating combs. We have monitored the volatile emissions from olive fruits stored for 5 consecutive days between harvesting and processing (following the IGP and PDO rules). Although a good deal has been written in the literature related to aroma compounds from olive oil, little is known about the composition of volatile compounds from olive fruit, especially how the VOCs change over time. It is known, for example, that many compounds contribute to the volatile composition of olive fruits, and variability among different cultivars has also been highlighted both in leaves and fruits [25,38,39]. Hence, after obtaining a general overview of volatile emissions from the two cultivars (Figure 2), we focused our attention on those compounds that showed an increasing trend over time. With our VOC analyses of olive fruits, 32 different signals were detected at each sampling time (T1–T5), as reported in Appendix A. All the signals were detected between protonated m/z 20 and m/z 101, and the same signals, characterized by different intensities, were identified in each cultivar and for each sampling time. 

Moreover, since the VOC total emissions trend in the stored samples (5 °C) of both cultivars tended to be remarkably similar in the two tested seasons, we considered only the data collected in 2019 to show the results better. From the data reported in Figure 2, we observed how the total VOC emissions (excluding from the counts the protonated m/z 33, 45 and 47, which had a contrary trend) from olives stored at 5 and 18 °C, respectively, are always higher at each sampling times in the “Frantoio” olives compared to the “Moraiolo”. On the contrary, in the “Moraiolo” olives, the VOC emissions decrease over time less markedly than in the “Frantoio” olives. Thus, during the time of storage, a decrease in the total VOC emissions (mostly linked to emission reduction of C5 and C6 compounds) emerged by one side, while an increase in emissions of compounds detected at m/z 33.033 (identified as methanol), m/z 45.033 (identified as acetaldehyde) and m/z 47.049 (identified as ethanol) by the other side (Figure 2 and Figure 3). In particular, the increase in differences began at T3 and continued between the fourth and fifth sampling times (Figure 3), when we noticed an emission increase in the compounds linked to oxidative and fermentative processes (ethanol, methanol and acetaldehyde increased noticeably). These compounds tend to increase during storage. 

Indeed, Ueda and co-authors (2019) [40] reported that in other fruits (e.g., tomatoes and strawberries), the ethanol and methanol emissions increased after cutting and following the storage time. In addition, Beltran et al. (2021) [41] reported that ethanol and acetaldehyde content increased inside the olive fruits after storage. Although ethanol and methanol are fundamental compounds for the aroma of fruits when they exceed certain thresholds of perception, they change into unpleasant odors [40]. It is known that methanol and ethanol, which are compound precursors of ethyl and methyl esters, occur naturally at a low level in fresh fruits, as well as acetaldehyde, which is an ethanol precursor [40,42]. As reported by Beltran et al. (2021) [41], when olives are stored, the ethanol content increases inside the fruit, and this phenomenon is reflected in a deterioration in olive oil quality. It is worth noting that ethanol in olive fruit is one of the precursors of ethyl esters, which are a virgin olive oil quality parameter adopted recently by legislation to discriminate “extra virgin olive oil” produced from healthy and high-quality olive fruits [43]. This behavior was more restrained in the “Moraiolo” compared to the “Frantoio” olive fruits (Figure 3). The same trend was observed both on the olives stored at 5 °C and in the others stored at 18 °C, as well as on fruits mechanically or manually harvested. The changes are more marked both in the olives stored at 18 °C and in those harvested mechanically. To conclude, since the volatile compounds strongly influence the olive oil quality and its acceptance, we need more studies to better explain how the harvesting system and storage time affect and interact with olive VOCs and olive oil quality.

#### 3.4.2. Influence of Olive Cultivar and Time Storage on VOC Emissions in Olive Oil

By the headspace analysis of six experimental olive oil samples, 39 different signals were detected by the PTR-ToF-MS analysis (Appendix A). Among these VOCs, we focused on the compounds identified by Taiti et al. (2022) [26] in a previous paper as quality markers for distinguishing defective oils from true EVOO. In particular, Taiti et al. (2022) [26] highlighted the role of some key VOCs that can be used to successfully discriminate between “EVOO” and “non-EVOO” categories. Thus, the compounds detected at m/z 45.033 (TI: acetaldehyde), m/z 59.049 (TI: acetone), m/z 61.028 (TI: acetic acid/acetates) were identified as off-flavors (from oxidation or fermentation phenomena), while m/z 79.059, m/z 81.069 and m/z 99.080 (all signals associated with C5 and C6 molecules) were identified as positive flavors since they are the main VOCs (derived from the LipOXygenase) linked to a green-fruity odor note. 

Figure 4 shows the emissions intensity of the compounds linked to flavors and off-flavors in each realized experimental oil. Overall, among the analyzed samples, the highest values for positive attributes as well as the lowest values for negative attributes were observed in the samples obtained at T1 and evaluated as not defective by the panel. Subsequently, we found a strong reduction of C5 and C6 compounds (i.e., m/z 79.059, 81.069, and 99.080) emissions and an increase in compounds linked to sensorial defects (m/z 45.033, 59.049, and 61.028) in both the oils produced after olive storage (T5) compared to the oils produced at T1. As expected, this behavior was more pronounced in the samples stored at 18 °C compared to others stored at 5 °C, which have a lower intensity of emissions of the compounds linked to off-flavors. Thus, although the samples obtained with olives stored at 18 °C showed a good emissions level of C6 compounds, the level emissions of compounds linked to off-flavors is present in quantities higher than the perception threshold of panelists who identified them as defective through the Panel (Table 8). In the same way, the samples obtained through a mechanical harvesting system have higher emissions of the compounds linked to the defects than those harvested manually.

### 3.5. Sensorial Analysis

In the present work, a total of six samples obtained in the 2019/20 season were evaluated in a single sensorial session by trained panelists after 3 months from their mill separation. Assessors were asked to evaluate attributes such as: “aroma intensity”, “bitter” and “pungent” characters and to evaluate the presence of perceptible off-flavors (splitting them into oxidations and fermentations). Figure 5 shows the results of the panel test for the samples stored at 5 and 18 °C obtained from a blend of the two cultivars. Sensorial differences were observed between the oil separated from olives stored at 5 °C and those stored at 18 °C (T5). Indeed, after the olive storage (T5), the panelists observed a decrease in aroma intensity, with “pungent” and “bitter” characters compared to the samples obtained at T1 (Figure 5). In addition, the off-flavors were highlighted by the panelists inside the samples stored at room temperature, mostly linked to fermentation and oxidation processes, and therefore, they were identified as defective samples. The oil samples obtained from olives harvested manually had a little less “aroma intensity” and a minor “pungent” note compared to the mechanically harvested samples. Our results agree with Brkić Bubola et al. (2020) [44], wherein after the fruit storage of cv Rosinjola, the panelists observed a decrease in aroma intensity (fruity), bitterness and pungent character. Based on sensory analysis, it seems that low-temperature storage (5 °C) of olive fruits may help maintain the high sensory quality of oils. On the contrary, the mechanical compared to manual harvesting seems to not affect the final oil quality, at least at T1, while it seems to determine differences over the long storage period.

### 3.6. Influence of Olive Cultivar and Time Storage on Olive Oil Quality Parameter

Table 8 reports changes observed in the olive oil samples both due to the different harvesting and storage systems. No difference was found both for free acidity (FA) and peroxide index (PI) in the experimental oil samples obtained with a different harvesting system at T1, while differences were observed at T5. At T5, we observed differences linked both to the harvesting system and storage temperature (Table 8). For example, after five days of storage, the olives harvested manually showed FA values ranging from 0.18% (T1) to 0.24% or 0.30% if stored at 5 or 18 °C, respectively; on the contrary in hand-held machine-harvested olives, the FA ranged from 0.20% (T1) to 0.27% if stored at 5 °C and up to 0.45% if stored at 18 °C. The PI had a similar trend, namely a slower increase in olives harvested manually with respect to those mechanically harvested. These results confirm what was previously reported by Famiani et al. (2020) [45], which found that the FA and PI increase more slowly during storage when the olives are harvested manually. Thus, the oil samples obtained by the mechanically harvested olives showed higher average values than those harvested manually. The samples stored at 18 °C showed the lowest quality level compared to the refrigerated ones. However, no sample exceeded both the acidity limit threshold (0.8%), and the peroxide values limit established (20 meq O_2_ kg^−1^) for the extra virgin olive oil by European rules (Table 8).

Regarding the total phenolic compounds, we observed a decrease over time that was linked to the applied storage temperature and harvesting system. The decrease in total phenolic compounds was more noticeable in the samples stored at 18 °C and harvested mechanically (Table 8). Total polyphenol in refrigerated samples decreased from 420 mg kg^−1^ at T1 to 380 mg kg^−1^ at T5, while the non-refrigerated samples decreased from 420 mg kg^−1^ to 315 mg kg^−1^ at T5. The reduction in total phenol content linked to storage temperature agrees with the results previously reported by other authors [46,47]. Thus, it emerges that the storage of olives at 5 °C helps keep high levels of polyphenols in the EVOO. This effect is important for the quality of EVOO because a high polyphenol content contributes to the flavor and guarantees a long shelf-life [48]. In sum, a storage temperature of 5 °C had a positive effect on the FA, PI and polyphenol contents regardless of the harvesting system. However, the olive oils obtained at T5 with olive stored at 5 °C showed a lower quality compared to the oil samples obtained at T1. Moreover, although the mechanically harvested olives did not show external injuries, it seems that this harvesting system may reduce the final olive oil quality when obtained from olives stored for more days. The chemical results obtained in this study confirm those previously reported by Yousfi and co-authors [37,47] in terms of cold storage. Thus, when it is not possible to guarantee that refrigerated olives will be milled in the near future, the use of climatic chambers appears to be indispensable to safeguard the quality of the extra virgin olive oils.

## 4. Conclusions

The data show how mechanical harvesting carried out by electric vibrating combs increases bruising damage compared to manual harvesting. Moreover, the bruising level increases steadily over time during fruit storage. Nonetheless, when the olives (also collected mechanically) are kept at an appropriate storage temperature, the postharvest life could be prolonged without compromising the future olive oil quality. During storage, the primary fruit changes are represented by an increase in ethanol and methanol emissions, as well as a decrease in C5 and C6 compounds and a slight reduction in polyphenol content. Our results highlight how the storage temperature of 5 °C could slow the olive degeneration processes between the harvesting and the milling, maintaining the final olive oil quality. Therefore, storage at a temperature of 5 °C allows the oils to maintain their EVOO characteristics. By the data obtained, it seems that the major differences are linked to the length of the storage period rather than to the system of harvesting, even if, in the case of mechanical harvesting, the processes are more evident. Therefore, it is possible to find a relationship between the VOCs emitted by the olive fruits and the obtained oils; this information would make it possible to predict the time span for olive conservation; this time span could be increased by applying cold-storage techniques. Finally, as the storage time is regulated by the law, it seems important to underline that the prescribed storage times before milling should be reviewed in order to yield high-quality EVOO (PDO and PGI).

## Figures and Tables

**Figure 1 plants-12-03843-f001:**
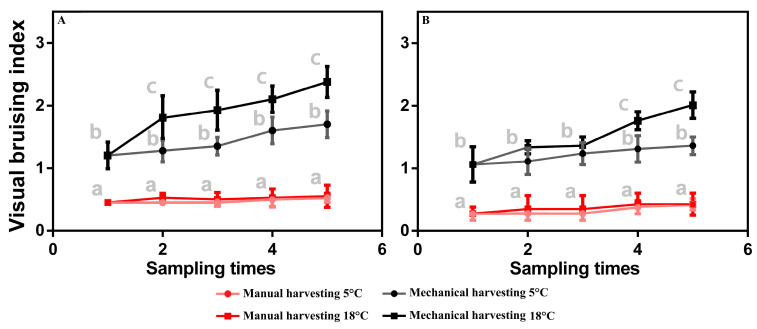
Visual Bruising Index for each sampling time (T1–T5) during postharvest conservation. (**A**) Values recorded for cv Frantoio; (**B**) values recorded for cv Moraiolo. Mean and standard deviation values are shown (n = 100). Different lowercase letters within the same sampling time indicate differences by the LSD test at the 95% confidence level (*p* = 0.05).

**Figure 2 plants-12-03843-f002:**
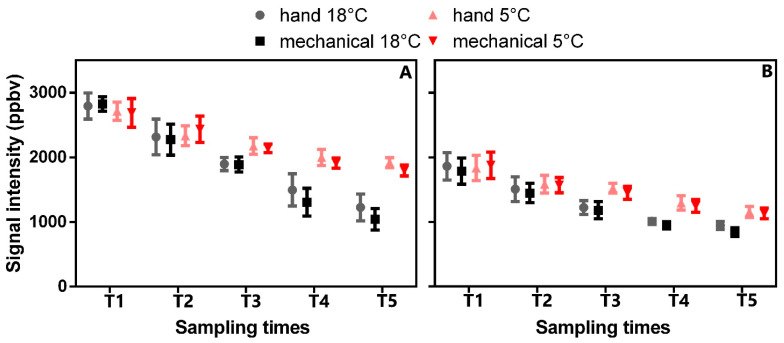
Trend evolution of total VOC (ppbv) emissions (excluding m/z 33.033, m/z 45.033, m/z 47.049) from olive fruits collected manually or mechanically by cultivars Frantoio (**A**) and Moraiolo (**B**) under storage after 1 (T1), 2 (T2), 3 (T3), 4 (T4), and 5 (T5) days after the harvesting. Mean and standard deviation values are shown (n = 100).

**Figure 3 plants-12-03843-f003:**
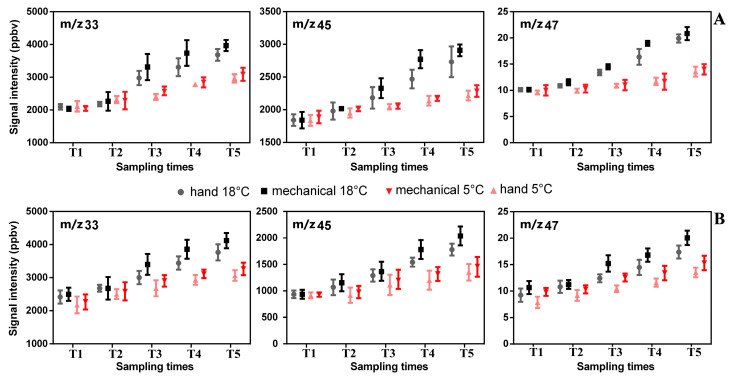
Means of signal intensities of some impact volatile compounds (m/z 33.033 Methanol, m/z 45.033 Acetaldehyde, m/z 47.049 Ethanol) detected in olive fruits under storage after 1 (T1), 2 (T2), 3 (T3), 4 (T4), and 5 (T5) days after the harvesting for cultivars Frantoio (**A**) and Moraiolo (**B**), collected manually and mechanically. Mean and standard deviation values are shown (n = 100).

**Figure 4 plants-12-03843-f004:**
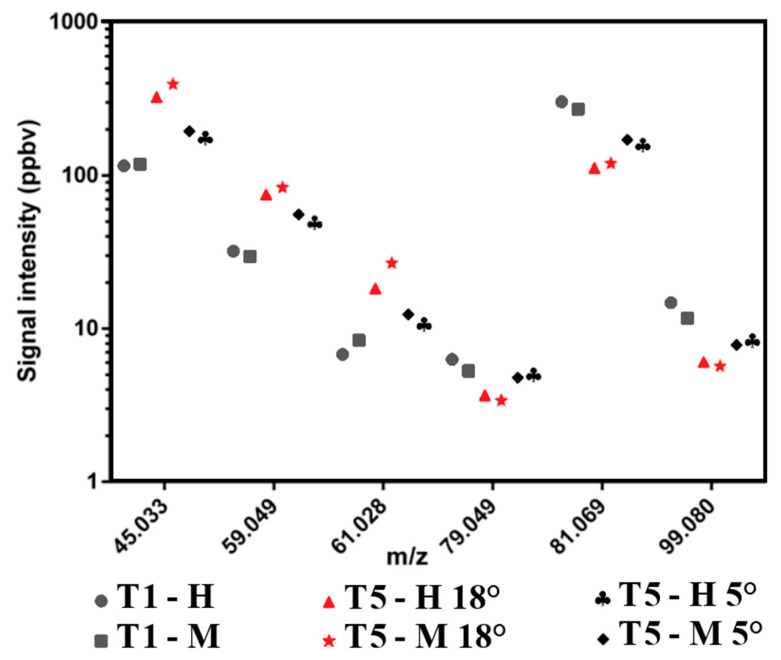
Means of signal intensities of some key VOCs that can be used to successfully discriminate “EVOO” and “non-EVOO” categories. For each oil sample studied (blend) are reported the compounds linked to flavors (m/z 79.049, m/z 81.069, m/z 99.080) and off-flavors (m/z 45.033, m/z 59.049, m/z 61.028). H—olive handmade harvested; M—olive mechanically harvested; T1—first sampling; T5—last sampling. Storage temperature: 18 °C and 5 °C.

**Figure 5 plants-12-03843-f005:**
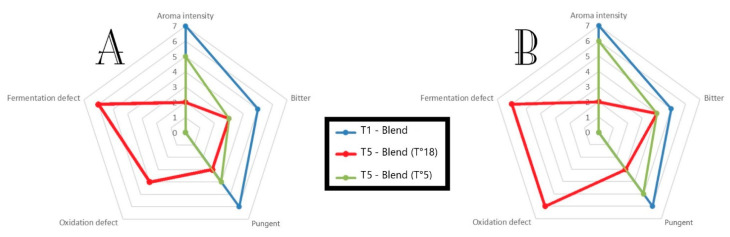
Effect of temperature storage and harvesting system on sensory quality of olive oil. (**A**) Samples obtained from manual harvesting; (**B**) samples obtained from mechanical harvesting.

**Table 1 plants-12-03843-t001:** Hours of olive storage allowed based on the different rules applied for 50 Italian high-quality EVOO production types (Italian IGP and DOP). * The storage time and/or the deadline for olive harvesting is not specified by law.

Number of Italian PGI and DOP	EVOO Quality Label	District	Allowed Time for Olive Storage (h)	Deadline for Olive Harvesting
1	DOP “Petruziano colline Teramane”	Abruzzo	48	10 December
2	DOP “Aprutino pescarese”	Abruzzo	72	10 December
3	DOP “Colline Teatine”	Abruzzo	*	20 December
4	IGP “Olio Lucano”	Basilicata	48	31 January
5	DOP “Vulture”	Basilicata	24	31 December
6	IGP “Olio di Calabria”	Calabria	24	15 January
7	DOP “Alto Crotonese”	Calabria	48	31 December
8	DOP “Bruzio”	Calabria	48	31 December
9	DOP “Lametia”	Calabria	*	15 January
10	DOP “Cilento”	Campania	48	31 December
11	DOP “Colline salernitane”	Campania	48	31 December
12	Dop “Irpinia Colline dell’Ufita”	Campania	48	31 December
13	DOP “Penisola Sorrentina”	Campania	48	31 December
14	DOP “Terre Aurunche”	Campania	48	31 December
15	DOP “Brisighella”	Emilia Romagna	48	20 December
16	DOP “Colline di Romagna”	Emilia Romagna	48	15 December
17	DOP “Tergeste”	Friuli	72	31 December
18	DOP “Canino”	Lazio	36	31 December
19	DOP “Colline pontine”	Lazio	48	31 January
20	DOP “Sabina”	Lazio	*	31 January
21	DOP “Tuscia”	Lazio	24	15 January
22	DOP “Riviera ligure”	Liguria	*	31 March
23	DOP Garda	Lombardia/Veneto/Trentino	120	15 January
24	DOP “Laghi Lombardi”	Lombardia	*	15 January
25	IGP “Marche”	Marche	*	15 December
26	DOP “Cartoceto”	Marche	48	25 December
27	DOP “Molise”	Molise	48	*
28	IGP “Olio di Puglia”	Apulia	36	31 January
29	DOP “Dauno Gargano”	Apulia	72	31 January
30	D.O.P. “Collina di Brindisi”	Apulia	*	31 January
31	DOP “Terre di Bari”	Apulia	*	31 January
32	DOP “Terre d’Otranto”	Apulia	48	31 January
33	DOP “Terre Tarentine”	Apulia	72	*
34	DOP “Sardegna”	Sardegna	48	31 January
35	IGP “Sicilia”	Sicily	48	31 January
36	DOP “Monti Iblei”	Sicily	48	*
37	DOP “Monte Etna”	Sicily	48	*
38	DOP “Valdemone”	Sicily	48	31 January
39	DOP “Val di Mazara”	Sicily	48	31 December
40	DOP “Valle del Belice”	Sicily	48	31 December
41	DOP “Valli Trapanesi”	Sicily	48	31 December
42	IGP “Toscana”	Tuscany	*	*
43	Dop “Chianti Classico”	Tuscany	72	*
44	DOP “Lucca”	Tuscany	48	31 December
45	DOP “Seggiano”	Tuscany	48	15 January
46	DOP “Terre di Siena”	Tuscany	72	31 December
47	DOP “Umbria”	Umbria	*	15 January
48	DOP “Veneto Valpolicella”	Veneto	*	15 January
49	DOP “Veneto Euganei e Berici”	Veneto	*	15 January
50	DOP “Veneto del Grappa”	Veneto	*	15 January

**Table 2 plants-12-03843-t002:** Total rainfall (mm), rainy days (n°) and temperatures (°C) measured through a weather station located inside the olive orchard for two consecutive years (2018–2019).

	Total Rainfall (mm)		Rainy Days (n°)		Average Temperature (°C)	
	2018	2019	2018	2019	2018	2019
Jan	66.8	32.4	13	5	9.2	5.1
Feb	112	72.6	10	5	6.1	9.6
Mar	201.2	7.2	16	4	14.4	12
Apr	58.4	88	8	11	16.7	13.7
May	74.2	136.4	8	14	19.3	15.4
Jun	39.2	18.2	4	2	22.4	24.5
Jul	45.4	37.6	4	4	25.9	26.3
Aug	71.6	14.2	5	3	25.8	26.6
Sep	9.8	105.4	1	9	22.6	21.7
Oct	70.6	44.6	6	5	18	17.5
Nov	69.6	327.8	13	22	12.2	12.5
Dec	73.6	138.8	9	11	7.3	9.2
Total	892.4	1023.2	97	95		
Average					16.7	16.2

**Table 3 plants-12-03843-t003:** Average of Fresh Weight (FW), Resistance to Detachment (RD) and Ripening Index (RI) evaluated on fruits (n = 200) collected from cv Frantoio and Moraiolo in two different seasons (2018/19, year-off and 2019/20, year-on).

Year	Alternate Bearing	Cultivar	FW (g) ± DS	RD (g)	RI
2018/19	OFF	Frantoio	2.90 ± 0.66	460	3.2
		Moraiolo	2.50 ± 0.45	480	2.9
2019/20	ON	Frantoio	2.70 ± 0.35	500	2.3
		Moraiolo	1.80 ± 0.28	580	1.9

**Table 4 plants-12-03843-t004:** Summary of Multifactor ANOVA results for Visual Bruising Index obtained using a storage temperature of 5 °C; factors: harvesting system—HS (manual vs. mechanical); sampling time—T (T1–T5); cultivar—cv (Frantoio vs. Moraiolo); year—Y (2018 vs. 2019).

Source	Sum of Squares	Effect (%)	Df	Mean Square	F-Ratio	*p*-Value
Main Effects						
HS	15.62	42.90	1	15.62	566.43	0.000
T	2.92	8.00	4	0.73	26.46	0.000
cv	8.70	23.90	1	8.70	315.19	0.000
Y	3.67	10.00	1	3.60	131.97	0.000
Interactions						
HS ×T	0.07	-	4	0.02	0.66	0.620
HS × cv	0.90	2.40	1	0.90	32.05	0.000
HS × Y	1.96	5.40	1	1.95	70.72	0.000
T × cv	0.01	-	4	0.00	0.13	0.969
T × Y	0.05	-	4	0.01	0.46	0.767
cv × Y	0.02	-	1	0.02	0.68	0.412
HS × T × cv	0.16	-	4	0.04	1.46	0.221
HS × T × Y	0.01	-	4	0.00	0.12	0.975
HS × cv × Y	0.10	-	1	0.10	3.70	0.057
T × cv × Y	0.02	-	4	0.01	0.21	0.931
HS × T × cv × Y	0.03	-	4	0.01	0.27	0.895
Residual	2.21		80	0.03		
Total (Corrected)	36.41	100	119			

All F-ratios are based on the residual mean square error.

**Table 5 plants-12-03843-t005:** Summary of Multifactor ANOVA results for Visual Bruising Index, the year 2019; factors: harvesting system—HS (manual vs. mechanical); sampling time—T (T1–T5); cultivar—cv (Frantoio vs. Moraiolo); storage temperature—ST (5 °C vs. 18 °C).

Source	Sum of Squares	Effect (%)	Df	Mean Square	F-Ratio	*p*-Value
Main Effects						
HS	4.96	26.40	1	4.96	168.66	0.0000
T	3.15	16.70	4	0.79	26.77	0.0000
cv	5.98	31.80	1	5.98	203.47	0.0000
ST	0.41	2.20	1	0.41	13.88	0.0004
Interactions						
HS × T	0.02	-	4	0.05	0.19	0.9446
HS × cv	0.67	3.30	1	0.62	20.95	0.0000
HS × Y	0.11	-	1	0.11	3.67	0.0589
T × cv	0.32	1.70	4	0.08	2.73	0.0347
T × Y	0.23	-	4	0.06	1.98	0.1050
cv × Y	0.13	0.70	1	0.13	4.53	0.0363
HS × T × cv	0.04	-	4	0.01	0.34	0.8483
HS × T × Y	0.02	-	4	0.00	0.14	0.9649
HS × cv × Y	0.03	-	1	0.03	0.92	0.3409
T × cv × Y	0.31	-	4	0.08	2.68	0.0376
HS × T × cv × Y	0.14	-	4	0.04	1.23	0.3050
Residual	2.35	12.50	80	0.03		
Total (Corrected)	18.84	100	119			

All F-ratios are based on the residual mean square error.

**Table 6 plants-12-03843-t006:** Influence of harvesting system and storage conditions on total phenolic content of cultivars Frantoio and Moraiolo collected in 2019. Data are reported as mg GAE/g ss ± SD.

Year	Alternate Bearing	Storage T°	Cultivar	Harvesting System	T1 *		T3 *		T4 *	
					Average	SD	Average	SD	Average	SD
2019/2020	OFF	18 °C	Frantoio	manual	19.8	0.6	23.1	0.8	21.3	0.5
				mechanical	19.9	0.8	20.7	2.1	22.8	3.0
			Moraiolo	manual	33.5	0.6	30.5	0.3	27.8	1.4
				mechanical	29.3	0.9	29.9	1.2	28.2	1.1
		5 °C	Frantoio	manual	26.5	1.3	24.2	0.8	24.4.	0.8
				mechanical	21.2	1.3	22.7	1.6	22.0	1.0
			Moraiolo	manual	43.8	1.8	43.3	1.2	41.2	0.6
				mechanical	40.0	0.5	40.6	0.7	39.6	0.8

* Sampling times at 1 (T1), 3 (T3), and 4 (T4) days after the harvesting, respectively.

**Table 7 plants-12-03843-t007:** Influence of harvesting system and storage conditions on Hydroxytyrosol (OHTyr), Oleuropein (Ole), Rutin, and Verbascoside (Verb) content of cultivars Frantoio and Moraiolo collected during the year-on (2019). Data are reported as mg/g ss ± SD.

			18 °C				5 °C			
	Harvesting System	Sampling Times (*)	OHTyr	Ole	Rutin	Verb	OHTyr	Ole	Rutin	Verb
Frantoio	manual	T1	0.13 ± 0.03	0.07 ± 0.00	0.26 ± 0.03	2.09 ± 0.38	0.14 ± 0.00	0.09 ± 0.02	0.58 ± 0.00	2.63 ± 0.12
		T4	0.30 ± 0.04	0.26 ± 0.01	0.41 ± 0.02	1.73 ± 0.23	0.11 ± 0.00	0.17 ± 0.06	0.56 ± 0.18	2.23 ± 0.35
	mechanical	T1	0.17 ± 0.02	0.11 ± 0.01	0.56 ± 0.06	2.19 ± 0.20	0.09 ± 0.03	0.07 ± 0.03	0.38 ± 0.05	1.14 ± 0.00
		T4	0.31 ± 0.01	0.33 ± 0.13	0.56 ± 0.07	0.70 ± 0.02	0.08 ± 0.00	0.09 ± 0.06	0.57 ± 0.21	1.61 ± 0.17
Moraiolo	manual	T1	0.14 ± 0.02	0.16 ± 0.09	0.75 ± 0.00	3.53 ± 0.82	0.19 ± 0.02	0.78 ± 0.03	1.30 ± 0.18	9.09 ± 0.97
		T4	0.30 ± 0.02	0.86 ± 0.11	0.85 ± 0.16	2.25 ± 0.07	0.14 ± 0.02	1.03 ± 0.54	1.54 ± 0.24	7.03 ± 1.3
	mechanical	T1	0.20 ± 0.03	0.61 ± 0.37	0.88 ± 0.38	3.61 ± 0.73	0.17 ± 0.01	0.60 ± 0.15	1.31 ± 0.04	6.64 ± 0.28
		T4	0.33 ± 0.00	1.04 ± 0.14	0.97 ± 0.11	1.27 ± 0.03	0.18 ± 0.01	1.51 ± 0.01	1.50 ± 0.12	4.93 ± 0.57

(*) Sampling times at 1 (T1) and 4 (T4) days after the harvesting, respectively.

**Table 8 plants-12-03843-t008:** Changes in FA (free acidity, %), PI (peroxide index, meq O_2_ kg^−1^), total polyphenols (mg kg^−1^) during the oil storage (mean values for all studied samples) and Sensory Evaluation results. To be marketed as extra virgin olive oil, samples must have FA < 0.8 and PI < 20 without off-flavors perceived by SE.

Storage T (°C)	18		5		18	
Sampling Times	T1		T5		T5	
Harvesting system	Manual	Mechanical	Manual	Mechanical	Manual	Mechanical
Free acidity (%)	0.18 ± 0.01	0.20 ± 0.02	0.24 ± 0.03	0.27 ± 0.04	0.30 ± 0.03	0.45 ± 0.03
Peroxide index (meq O_2_ kg^−1^)	5.70 ± 0.30	5.60 ± 0.35	6.90 ± 0.60	9.20 ± 0.70	10.20 ± 0.80	13.50 ± 0.90
Total polyphenols (mg kg^−1^)	420 ± 14	417 ± 16	411 ± 12	381 ± 18	372 ± 15	315 ± 12
SE results (defective, not defective)	not defective	not defective	not defective	not defective	defective	defective

## Data Availability

Data are contained within the article and Appendix A.

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
