# Peer review of "Does the Harvest Type Affect Olive Health? Influence of the Harvesting System and Storage Time on the Chemical, Volatile and Sensory Qualities of Extra Virgin Olive Oils"

_plants, 2023, doi:10.3390/plants12223843_

Round 1

Reviewer 1 Report

Comments and Suggestions for Authors

In this manuscript (plants-2721163) entitled "Does the harvest type affect the olives health? Influence of the harvesting system and storage time on chemical, volatile and sensory quality of extra virgin olive oils" submitted to Plants, Cosimo Taiti and colleagues have investigated the effect of bruising and its development during the postharvest time on the olive fruits (cv Frantoio and Moraiolo) under two different temperatures. This research is interesting and convincing, but minor points need to be addressed to improve the quality of this manuscript.

1. To better understand this study, harvesting and storage pictures of olive fruits (cv Frantoio and Moraiolo) should be shown in the revised manuscript. In addition, sampling, harvesting and storage locations should be marked on a map in the revised Figure.

2. For Figure 1, too many data are shown in a confusing way in this Figure. Authors should consider to show the total rainfalls (mm), rainy days (n°) and temperatures (°C) as separate panels in the revision.

3. Data presented in this manuscript was mainly collected during two consecutive years (2018-2019). What is the situation for the recent four years (2020-2023)?

4. For Figure 2, Authors should consider to employ four different colors to show the visual bruising index for manually and mechanically harvesting at two different temperatures (5°C and 18°C). Same concerns applied to Figure 3 and 4.

5. For Table 4, full names of abbreviations presented in this table should be spelt out in the revised manuscript.

Author Response

We thank the reviewer for his/her positive comments on our work. We agree with the referee and we have carefully edited the manuscript to improve its structure and revise the terminology throughout the text.

To better explain our study, we added on the graphical abstract the indication about the orchard location.

Figure 1 has been converted into a table to better understand the data shown. The Figure 2 was corrected according to the reviewers suggestions (we used four different colors).  

The climate data resulted similar even if the 2022/2023 are characterize by extreme climate events: summer and  autumn very hot and dry.

Table 4 was corrected as suggested by the reviewer. 

Reviewer 2 Report

Comments and Suggestions for Authors

This research evaluates the impact of mechanical harvesting vs manual harvesting an the storage temperature of olive fruits on the quality of the oil produced.

The introduction is complent and presents revelant data. It also present the problem correctly.

The material and methods are complete and offers a large panel of methods tested to evaluate the modifications in the quality.

The results and discussion is complete.  With a coherent conclusion.

There are several minor things that should be corrected along the document to make it more clear. There are some English errors. Please where you should made the corrections in the attached file

Comments on the Quality of English Language

The are some errors and my english level do not allow to check/correct perfectly

Author Response

We are pleased that the reviewer found that our paper delivers novel findings. We agree with the referee and we have corrected the manuscript based on their suggestion.